# Enabling interpretable machine learning for biological data with reliability scores

K. D. Ahlquist[1,2]*, Lauren A. Sugden[3‡]*, Sohini Ramachandran[1,4,5‡]

**1** Center for Computational Molecular Biology, Brown University, Providence, Rhode Island, United States of America, **2** Department of Molecular Biology, Cell Biology, and Biochemistry, Brown University, Providence, Rhode Island, United States of America, **3** Department of Mathematics and Computer Science, Duquesne University, Pittsburgh, Pennsylvania, United States of America, **4** Department of Ecology, Evolution and Organismal Biology, Brown University, Providence, Rhode Island, United States of America, **5** Data Science Initiative, Brown University, Providence, Rhode Island, United States of America

‡ These authors are joint senior authors on this work.
* kaileigh_ahlquist@alumni.brown.edu (KDA); sugdenl@duq.edu (LAS)

**Data Availability Statement:** All relevant data are within the manuscript and its Supporting Information files. Data and code are available on Github: https://github.com/ramachandran-lab/SRS_paper

## Abstract

Machine learning tools have proven useful across biological disciplines, allowing researchers to draw conclusions from large datasets, and opening up new opportunities for interpreting complex and heterogeneous biological data. Alongside the rapid growth of machine learning, there have also been growing pains: some models that appear to perform well have later been revealed to rely on features of the data that are artifactual or biased; this feeds into the general criticism that machine learning models are designed to optimize model performance over the creation of new biological insights. A natural question arises: how do we develop machine learning models that are inherently interpretable or explainable? In this manuscript, we describe the SWIF(r) reliability score (SRS), a method building on the SWIF(r) generative framework that reflects the trustworthiness of the classification of a specific instance. The concept of the reliability score has the potential to generalize to other machine learning methods. We demonstrate the utility of the SRS when faced with common challenges in machine learning including: 1) an unknown class present in testing data that was not present in training data, 2) systemic mismatch between training and testing data, and 3) instances of testing data that have missing values for some attributes. We explore these applications of the SRS using a range of biological datasets, from agricultural data on seed morphology, to 22 quantitative traits in the UK Biobank, and population genetic simulations and 1000 Genomes Project data. With each of these examples, we demonstrate how the SRS can allow researchers to interrogate their data and training approach thoroughly, and to pair their domain-specific knowledge with powerful machine-learning frameworks. We also compare the SRS to related tools for outlier and novelty detection, and find that it has comparable performance, with the advantage of being able to operate when some data are missing. The SRS, and the broader discussion of interpretable scientific machine learning, will aid researchers in the biological machine learning space as they seek to harness the power of machine learning without sacrificing rigor and biological insight.

**Funding:** This work was supported by the US National Institutes of Health R01GM118652 (KDA, LAS, SR) and R35GM139628 (KDA, SR), and the Wimmer Family Foundation (LAS). KDA was also supported as a trainee by NIH T32GM007601. The funders had no role in software design, data collection or analysis, decision to publish or the preparation of the manuscript.

**Competing interests:** The authors have declared that no competing interests exist.

## Author summary

Machine learning methods are incredibly powerful at performing tasks such as classification and clustering, but they also pose unique problems that can limit new insights. Complex machine learning models may reach conclusions that are difficult or impossible for researchers to understand after-the-fact, sometimes producing biased or meaningless results. It is therefore essential that researchers have tools that allow them to understand how machine learning tools reach their conclusions, so that they can effectively design models. This paper builds on the machine learning method SWIF(r), originally designed to detect regions in the genome targeted by natural selection. Our new method, the SWIF(r) Reliability Score (SRS), can help researchers evaluate how trustworthy the prediction of a SWIF(r) model is when classifying a specific instance of data. We also show how SWIF(r) and the SRS can be used for biological problems outside the original scope of SWIF(r). We show that the SRS is helpful in situations where the data used to train the machine learning model fails to represent the testing data in some way. The SRS can be used across many different disciplines, and has unique properties for scientific machine learning research.

This is a *PLOS Computational Biology* Methods paper.

## Introduction

Interpretability has become an increasingly important area of research in machine learning [1,2], although the term "interpretability" is not always well-defined [3,4] When machine learning approaches are applied to biological problems, two major elements of interpretability are desirable: 1) the ability to connect results generated by machine learning applications with existing biological theory and understanding of biological mechanisms, and 2) the ability to identify and characterize the limitations of a given machine learning algorithm, so that it is applied correctly and appropriately in the biological context of interest. Recent field-specific papers have addressed the first aspect of biological interpretability, by identifying contexts where machine learning approaches are especially powerful, and identifying the types of insights that can be gained [5–9]. The second aspect, ensuring that machine learning is applied correctly and appropriately in the biological context, is also a prominent concern in the field, with recent studies suggesting standards for reporting machine learning results in a biological context [10–12], auditing practices for detecting bias in machine learning applications to biology [13], and guidelines for preventing common pitfalls in machine learning written expressly for biologists [14,15].

Ideally, machine learning workflows should include steps that evaluate the fit between training and testing data: testing for outliers or out-of-distribution instances, and distribution shifts between training data and testing data [16,17]. This has received particular attention in the context of neural networks [18–20], but in the context of other types of classifiers, tools available for the process of outlier detection or detecting distribution differences typically rely on different mathematical frameworks than the machine learning model and require an independent analysis. For example, one can detect outliers using the density of instances with the Local Outlier Factor [21], then subsequently apply an unrelated machine learning method to make classifications. Researchers have to rely on their own knowledge and discretion to decide which, if any, evaluations to include in their experiments.

Here, we introduce the SWIF(r) reliability score (SRS) with the goal of contributing to the goals of interpretability and best-practices in machine learning workflows. We seek to simplify

the decision about how to include testing for outliers and distribution differences in a few ways. By packaging our classifier and the SRS together, we prioritize and highlight the need for evaluations of outliers and distribution differences to be included in every machine learning analysis. Rather than performing outlier detection or looking for distribution differences prior to performing classification, or as a post-hoc analysis, researchers get SRS output and SWIF(r) classification output in a single step. Exclusion of outliers and many out-of-distribution instances can be performed by setting a threshold on the SRS, in much the same way that other methods for outlier and novelty detection can be applied. The SRS can also be used to measure distribution shifts, fitting into the "Failing Loudly" framework for machine learning described in Rabanser et al. 2019 [17], in which the authors advocate for machine learning tools that give warnings when facing unexpected input. In this particular framework, the SRS can serve in the role of dimensionality reduction, a step in the Failing Loudly pipeline prior to testing for significant distribution shift with subsequent two-sample tests.

We previously introduced SWIF(r) (SWeep Inference Framework (controlling for correlation), a supervised machine learning classifier that we applied to the problem of classifying genomic sites as neutrally evolving or undergoing positive selection in human population genetic data [22]. For this application, we show SWIF(r) to be competitive with other methods including deep neural networks and boosted logistic regression. SWIF(r) returns posterior class probabilities based on learned joint distributions of attributes (selection statistics in our motivating application) and user-provided priors on the class frequencies. The probabilities output by SWIF(r), and by other probabilistic classifiers, represent a posterior probability distribution across classes, with probabilities summing to 1 regardless of how well a classified instance matches the training data. As a consequence, a high classification probability can be untrustworthy for instances that do not resemble any of the classes included in training, while an outcome assigning equal probability to all classes may nevertheless be trustworthy if the classifier has been trained using many similar instances. When using a typical classifier, users will receive classification outcomes from their models, with little visibility into the underlying features contributing to classification [23,24], making the trustworthiness of individual classification decisions unclear; the purpose of the SRS is to provide this important feedback.

The use of selection statistics, or other immediately interpretable attributes, as features for training SWIF(r) contributes to the first aspect of interpretability described above: the ability to connect results generated by machine learning applications with existing biological theory and understanding of biological mechanisms. In its original application, the features used for machine learning with SWIF(r) are derived from existing mathematical theory in population genetics [25–30]. Working with a limited number of attributes that are highly meaningful in the biological context, whether summary statistics or other measurements of biological features, creates opportunities for biological interpretation—through the generation of testable hypotheses, or through critical dissection of the classification outcomes. SWIF(r) is able to perform classification even when instances contain missing attributes, an unusual feature in the machine learning space. While in some circumstances missing data can be imputed, SWIF(r) serves a particularly important role in circumstances when data cannot be reliably imputed, such as when data is missing in a meaningful pattern (aka. "missing not at random") [31,32], or when sample size and proportion of missing data restrict imputation [33,34].

The generative model underlying SWIF(r) also provides opportunities for interpretability of the second type: characterizing the limitations of the classifier in the context of a specific biological dataset. The SRS is derived from the Averaged One-Dependence Estimation (AODE) framework [35] underlying SWIF(r). An AODE is a generative classifier, learning the attribute distributions (or likelihoods) for each class from training data. Posterior distributions for classes that condition on observed attributes can then be derived and used to assign class

probabilities to new instances. This is in contrast to discriminative classifiers, which includes neural networks, support vector machines, random forest, and k-nearest neighbors algorithms, which learn decision boundaries between classes directly. The SRS uses the AODE framework underlying SWIF(r), and the attribute distributions learned by the classifier, to measure the trustworthiness of particular instances, which can be thought of as the similarity between a given instance and the training data as seen by the trained model. This allows a user to distinguish between a classification prediction that is made because the instance in question is a genuine match to one of the trained classes, and a prediction made because the classifier has simply found the "least bad" option. We demonstrate how the SWIF(r) and SRS framework can be extended to a wide range of biological problems.

The SRS is of particular value for biological applications where it is necessary to use simulated training data, a practice that is ubiquitous in population genetics, and extends to many evolutionary questions across different areas of biology [36–41]. Simulation studies also underlie other analyses of complex biological phenomena, such as cancer progression [42,43]. In many cases, researchers evaluate the performance of an algorithm or method on simulated "testing" data, before applying it to "application" data drawn from biological samples. The reliability of these models relies in part on the degree of similarity between simulated data and application data, but a high degree of similarity can be difficult to achieve despite the best efforts of researchers. When differences do arise, it may be desirable to set aside data instances with poor fit, allowing the classifier to effectively abstain when it encounters signatures it has not been trained to handle. For example, in the context of genome scans for selection, such unexpected signatures can arise in regions of the genome with idiosyncratic recombination or mutation profiles that are not captured in simulated training data, such as the MHC [44,45]. Designing abstention into classifiers should be considered a best practice [46], especially for biomedical data [47]. Abstention in the context of discriminative classifiers can be addressed with the addition of an "abstention class" in training data [48] or an "abstention option" included in the loss function [49]. In the context of a generative classifier, we assert that the addition of an abstention class or abstention option is not necessary; we can instead use the trained model itself (via the SRS) to identify instances where abstention is desirable without requiring the model to learn features associated with unreliable or confusing instances.

In this study, we assess the ability of the SRS to identify deficiencies affecting the classifications produced by SWIF(r) across different biological applications and contexts. When a classifier encounters data that is not well-matched to any trained class, its default behavior can be erratic, and at worst, deceptive, as we illustrate in our experiments. The addition of the SRS allows for identification of many types of performance problems, and offers potential remedies via the removal of outliers and measurement of distribution shifts using a subsequent two-sample test. More aggressive remedies include the addition of new classes and instances, or the training of additional classifiers to account for important structure identified in the data. In particular, we study performance in situations in which the following occur: 1) classes of data are missing in the training data but present in the application data, 2) there exists a systemic mismatch between training data and application data, 3) there are missing attribute values for specific instances undergoing classification.

## Results

In this study we introduce the SWIF(r) Reliability Score (SRS), and explore its utility for applications to biological data. We define the SRS, and use simulations to demonstrate the sensitivity of the SRS to underlying data features, including the distribution of individual attributes and the structure of correlation between attributes (Fig 1). Next, we examine and analyze a

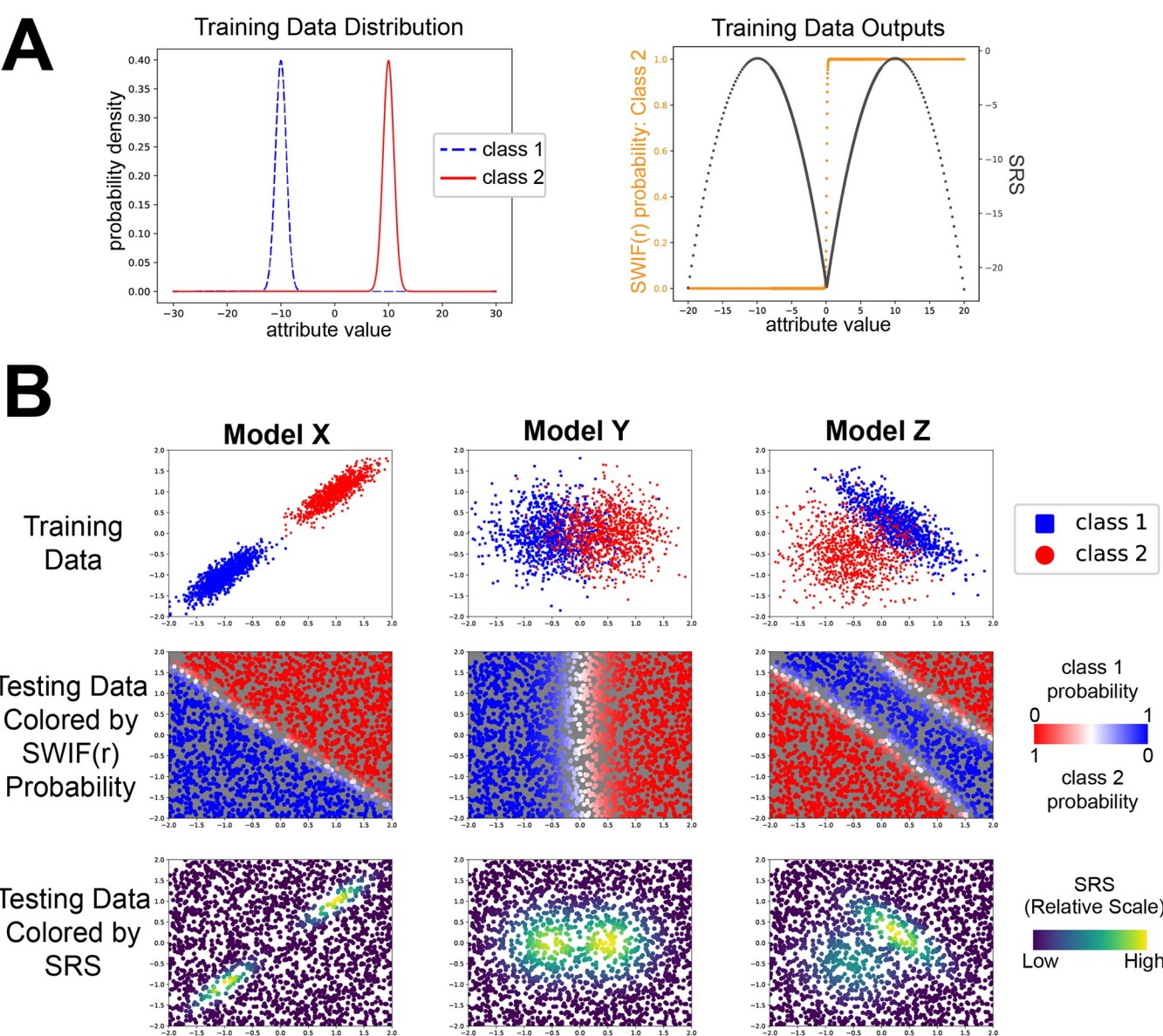

**Fig 1. The SRS reveals the underlying structure of the data in example training scenarios. A)** For a classification problem with a single attribute, we illustrate how the SRS rises and falls over a range of classification probabilities. Data was generated for two classes (class 1: blue, dashed line, and class 2: red, solid line) with a single attribute. SWIF(r) was trained using 1000 instances of each class drawn from the attribute distribution shown on the left. Following training, SWIF(r) was tested on values across the range (-20, 20), resulting in the SWIF(r) probabilities and SRS values shown on the right. In the right hand graph, SWIF(r) probability of class 2 is shown in orange, and SRS is shown in dark gray. Here we see that the SRS drops in regions that are not represented by either class in training, across a wide range of classification probabilities. **B)** The SRS detects differences in the correlation structure of testing data compared to training data in a two-attribute model. Data was generated for two classes (class 1 shown in blue, and class 2 shown in red) with two attributes. Different distributions and correlations between attributes were chosen to create three SWIF(r) models labeled "X" (left), "Y" (middle) and "Z" (right). In the top row for each model is the data used to train the model, with one attribute along each axis. The middle and bottom rows show testing data drawn uniformly across the same area shown. In the middle row, testing data is colored by the SWIF(r) probability, ranging from 0 to 1 for each class (note that for a binary classifier, $P(class\ 1) + P(class\ 2) = 1$). In the bottom row testing data is colored by the relative value of the SRS for the testing data shown, with yellow corresponding to the highest SRS, and purple corresponding to the lowest SRS. We note that all combinations of high and low SRS, and high and low SWIF(r) probabilities are possible, depending on the structure of the dataset and the particular instance being classified.

series of biological datasets to illustrate the utility of the SRS in applications to biological problems. First, we use a wheat morphology dataset to study the effect of a class that is present in testing data but unaccounted for in training data (Fig 2). Second, we use phenotype data

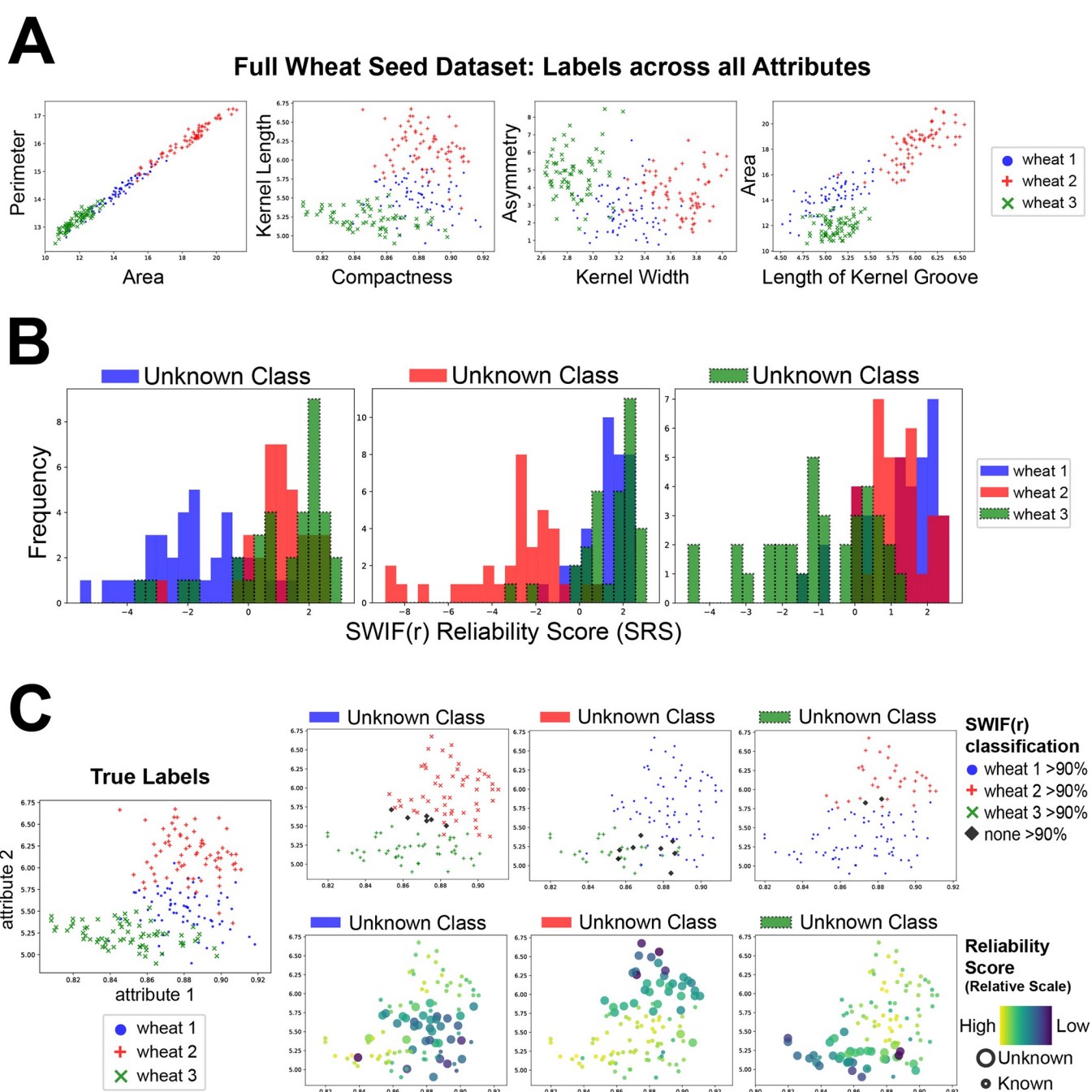

**Fig 2. The SRS is lower when an instance's class is excluded from training.** SWIF(r) was trained using each combination of two classes in the wheat morphology dataset [50] (see Methods). The trained model was then tested on instances from all three classes. **A)** Distribution of the training and testing data across all seven attributes: Perimeter, Area, Kernel Length, Compactness, Kernel Width, Asymmetry and Length of Kernel Groove. Note that wheat 1 (blue circles) generally has trait values that are intermediate to the other two classes. See S1 Fig for additional views into the raw dataset. **B)** Histograms show the distribution of the SRS for instances of each class. In each case, the unknown class has a more negative distribution of SRS compared to the two known classes. This is true even for wheat 1 (blue), which has attributes that are intermediate in value as compared to the other classes (see S1 Table for p-values). **C)** Data graphed by two attributes (Kernel Length and Compactness). Left: full data set colored by true labels. Right: colored by SWIF(r) probability (top) and SRS (bottom). SWIF(r) scores were generally greater than 90% for one of the two trained classes, *even for instances from the unknown class*, with just a handful of points receiving intermediate values from SWIF(r) (black diamonds). In contrast, coloring by SRS shows that points associated with the unknown class (larger dots) tend to have lower SRS, while points associated with known classes (smaller dots) received higher SRS.

drawn from the UK Biobank to study the effect of a systemic mismatch between training data and testing data, in this case due to differences in ancestry or sex across the training and testing sets (Fig 3). Third, we use simulated data and 1000 Genomes Project data to illustrate the effects of missing data on classification performance in a population genetics context (Fig 4).

## Overview and definition of the SRS

The SRS, developed here for SWIF(r) but generalizable to other generative classifiers, is an interpretability aid implemented from intermediate steps within SWIF(r)'s classification pipeline. SWIF(r) classifies instances using Averaged One-Dependence Estimation, an extension to Naive Bayes that incorporates pairwise joint distributions of attributes to estimate the joint likelihood of the observed data conditioned on each class. Briefly, from training data for each class, SWIF(r) learns a joint, two-dimensional Gaussian mixture for each pair of attributes $f(X_i = x_i, X_j = x_j | class)$, along with one-dimensional marginal Gaussian mixtures for each attribute $f(X_i = x_i | class)$. From the joint distributions, we derive conditional distributions $f(X_i = x_i | X_j = x_j, class)$. We then express the likelihood of the class given n attributes as follows:

$$L(class|X_1 = x_1, ..., X_n = x_n) \propto \sum_{j=1}^{n}[f(X_j = x_j|class)\prod_{i \neq j} f(X_i = x_i|X_j = x_j, class)] \qquad \text{(Eq1)}$$

Where $X_i$ represents the i$^{th}$ attribute and n is the number of attributes. With the use of prior class probabilities, and a normalization factor representing the marginal probability of the data summed over all classes, SWIF(r) returns a posterior probability for each class (see Materials and Methods, Eq 3). The SRS focuses solely on the estimated likelihoods, because these are the values that indicate the fit of the instance to the trained model, while the posterior probabilities in Eq 3 convey the *relative* fit across the trained classes. Because the posterior probabilities are forced to sum to 1, instances that do not resemble any trained class may still receive a high classification probability for the "least bad" option; the SRS is designed to help detect such instances.

To calculate the SRS, we calculate the estimated joint likelihoods given all n instances, take the maximum across all classes, then apply a $log_{10}$ transformation to aid visualization:

$$SRS = log_{10}(max_{class}(\sum_{j=1}^{n} f(X_j = x_j|class)\prod_{i \neq j} f(X_i = x_i|X_j = x_j, class))) \qquad \text{(Eq2)}$$

By taking the maximum value with respect to class, the SRS is agnostic to the specific class labels of an instance. Because the SRS depends heavily on the specifics of a particular application, SRS values cannot be compared across experiments. Instead, the scores are designed to be compared *within the context of a single trained model*.

In Fig 1, we illustrate how the SRS works in the context of two examples. In the first example (Fig 1A), classification into one of two classes is based on a single attribute, reducing the classification problem to a straightforward application of Bayes' rule. Observations in class 1 follow a Gaussian distribution with a mean of -1 and standard deviation 0.2, while observations in class 2 follow a Gaussian distribution with a mean of 1 and standard deviation 0.2. Note that in this case, the SRS reduces to the maximum value of the two probability density functions. As we classify values across the x-axis, we see that the probability assigned to class 2 monotonically increases, as we would expect. However, the SRS reports low reliability, or trustworthiness, in three places: values to the left of class 1, values between the two classes, and values to the right of class 2. In these ranges, we have values of testing data that are a poor match to both classes. We note in particular that low SRS values can occur across the full range

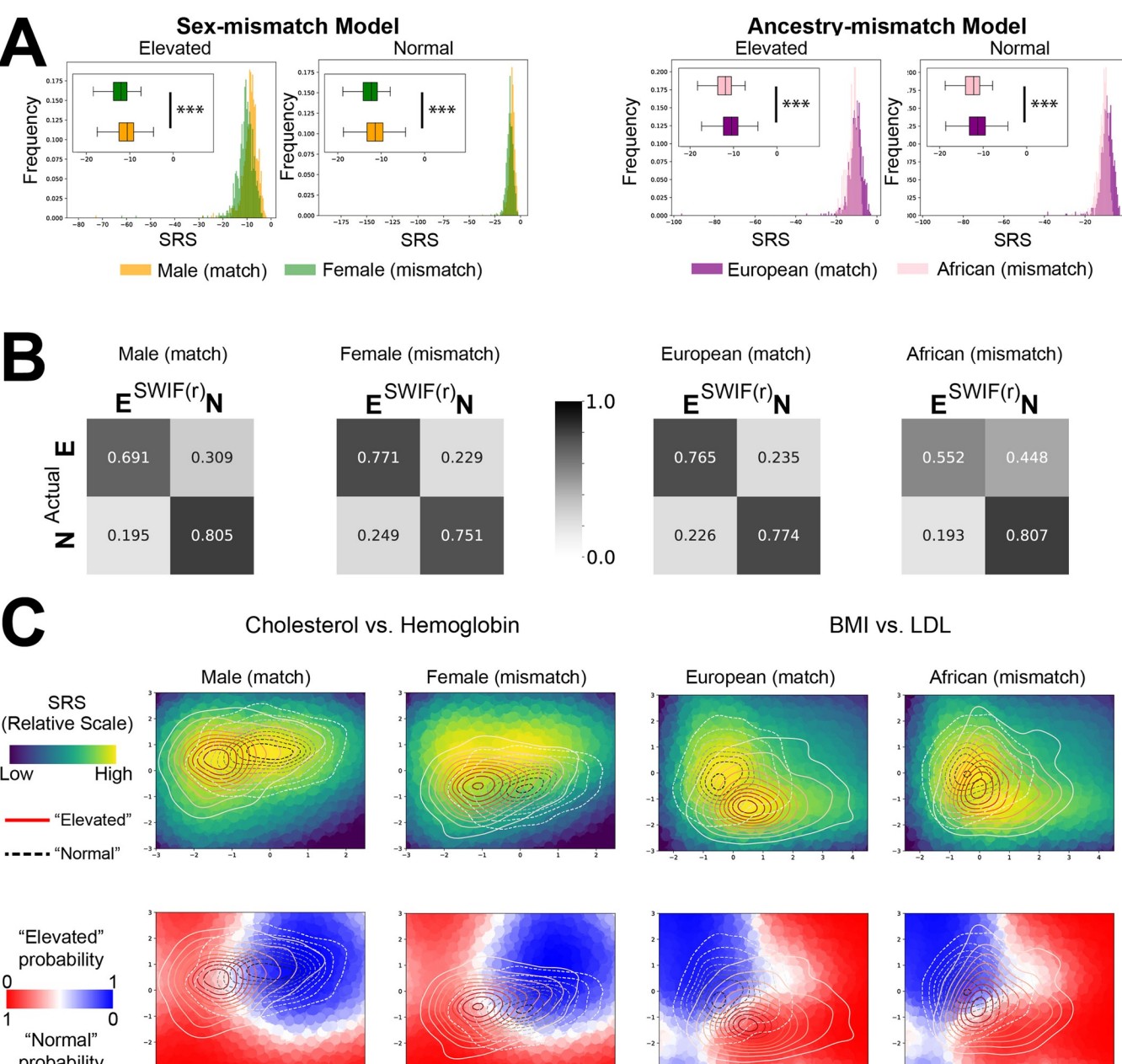

**Fig 3. Low average SRS can indicate systemic mismatch between training and testing data. A)** Histograms and boxplots showing the distributions of "match" and "mismatch" cohorts for each experiment. Histograms and boxplots show the same information, with boxplots zoomed in to allow for easier comparison of the distributions. The x-axis range in the histograms represents the complete range of observed SRS values in each case. In the sex-mismatch model (left), SWIF(r) was trained using a dataset consisting of male individuals of European ancestry divided into two categories based on HBA1C readings, used to esimate blood sugar: Elevated, or Normal. The data provided for training consisted of 22 health-related attributes (see Methods). The trained model was then tested on two cohorts, females with European ancestry and an independent cohort of males with European ancestry. These cohorts were labeled with their Elevated or Normal status, allowing for identification of correct or incorrect classification by SWIF(r). In the top left, we see the distribution of SRS for Elevated and Normal individuals from either the matching cohort (male) or non-matching cohort (female). The female cohort has lower average SRS, visible as a leftward shift in both the Elevated (t-test p-value = 1.53e-4) and Normal (t-test p-value = 5.11e-34) distributions (*** represents p<0.001). Likewise in the ancestry mismatch model (right), SWIF(r) was trained using a dataset of male and female individuals of European ancestry divided into two categories: Elevated and Normal. The trained model was then tested on two cohorts, males and females with African ancestry and an independent cohort of males and females with European ancestry. As above, we see the distribution of SRS for each cohort. The non-matching African ancestry cohort has lower average SRS, visible as a leftward shift in both the Elevated (t-test p-value = 6.04e-06) and Normal (t-test p-value = 1.50e-28) distributions. **B)** Confusion matrices show differences in SWIF(r) classification accuracy between matching and non-matching cohorts. The non-matching sex cohort experienced a small shift towards the Elevated classification when compared to the matching cohort. The non-matching ancestry cohort experienced a larger shift towards the Normal classification when compared to the matching cohort. **C)** SRS and SWIF(r) probability are calculated over a plane defined by two of the twenty-two model

attributes (Hemoglobin v Cholesterol for the sex-mismatch analysis, and BMI v LDL for the ancestry-mismatch analysis), providing a background of points for each graph. For other views into the data, see S6 and S7 Figs. On top of each is graphed a contour plot of the distribution of Elevated (red-to-white, solid line) or Normal (black-to-white, dashed line) data for each cohort. On the left, comparing the Male and Female cohorts we can observe a shift in the overall distribution of the data, pushing the Female cohort into an area with lower SRS values (top) and greater Elevated SWIF(r) probability (bottom). On the right, comparing the European and African cohorts, we observe that the mean difference between the Elevated and Normal cohorts is higher for individuals with European ancestry, and smaller for individuals with African ancestry. This results in greater overlap between the Elevated and Normal distributions in the African cohort, as well as an overall shift towards the Normal classification.

of classification probabilities, including probabilities that would be interpreted as "very likely class 1" to "very likely class 2" and everything in between. This makes the SRS a valuable, complementary addition to the standard classification output.

In Fig 1B, we scale up to examining a series of models with two attributes. The distribution of the training data is shown in the first row of panel B. The second row of panel B shows the SRS evaluated at points drawn uniformly across the plane. We see that the SRS is highest at

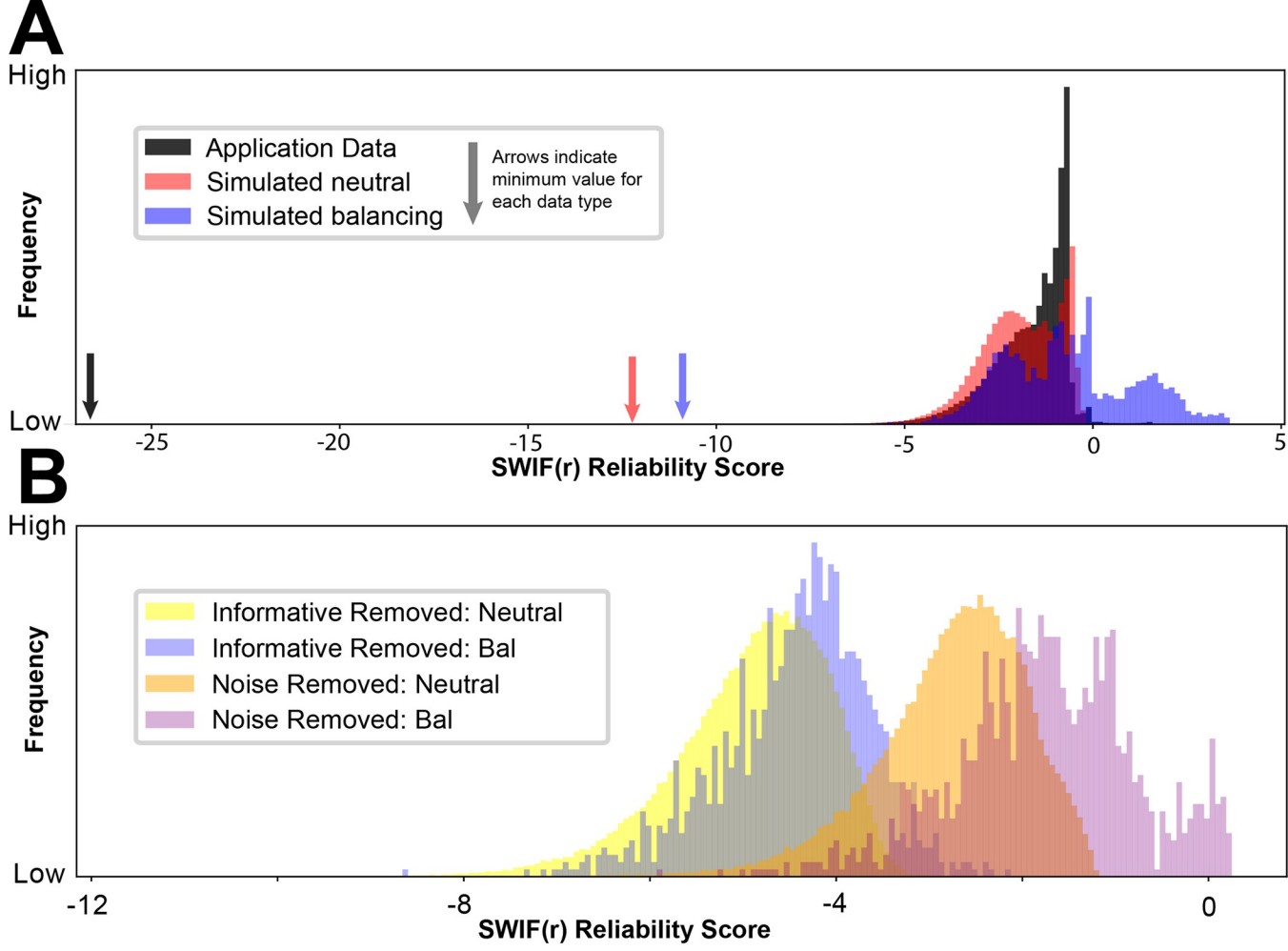

**Fig 4. The SRS can identify valuable attributes to create smarter filters on missingness. A)** Application data exhibits extreme low values of the SRS as compared to simulated data. **B)** Missing Informative attributes lead to lower SRS compared to missing Noise attributes. SWIF(r) was trained with a combination of five potentially informative attributes ("Informative"), and three "Noise" attributes drawn randomly from Gaussian distributions with no relation to the class of the instance. In the testing set, 3 attributes were removed from each instance: either all three Noise attributes ("Noise Removed"), or 3 randomly selected Informative attributes ("Informative Removed"). The SRS was then calculated for all instances. Instances where Informative attributes were removed have lower average SRS values compared to instances where the same number of Noise attributes were removed, visible in the histograms shown.

points similar to training data, not only in individual attribute value, but also with respect to the correlation between attributes. In particular, we see that for instances in which individual attribute values are all within typical ranges, the SRS will still report low values if the attributes jointly lie in a region of space not well-represented in the training data. This illustrates the sensitivity of the SRS to the high-dimensional structure of the trained dataset, which allows it to register an instance as "unexpected" even when no single attribute value constitutes an outlier with respect to its trained distribution. The bottom row shows the SWIF(r) probability evaluated at points drawn uniformly across the plane. Since SWIF(r) tries to find the "best" class to fit a particular instance, we see instances that lie far from the training data of both classes, as captured by the SRS, that are nevertheless classified with high probability to one class or the other. More generally, this illustrates the inherent limitations of a classifier as a set of decision boundaries. On the other hand, we can consider instances with SWIF(r) probabilities close to 0.5: in some cases, the SRS is low for these instances (Fig 1, panel B, left), while in other cases the SRS is high (Fig 1, panel B, center). For instances where SWIF(r) probability is close to 0.5 and the SRS is low, we have clearly separated distributions for each class, and points with SWIF(r) probability close to 0.5 are merely intermediate between the two classes (Fig 1, panel B, left) while being a poor fit to both. When SWIF(r) probability is close to 0.5 and SRS is high, this indicates that there is overlap in the distributions of the training data within which instances are well-fit by both classes, and are therefore ambiguous and difficult to classify (Fig 2, panel B, center). Combining SWIF(r) classification output with the SRS in this way enables a more complete and nuanced interpretation of results.

## The SRS is sensitive to instances from unknown classes

The SRS is sensitive to instances from classes in application data that are not present in the training data. While classifiers are generally forced to find the best match across trained classes, using the SRS allows researchers to identify those instances that are unusual with respect to all classes (Fig 1), and, potentially, to associate them with a class that was excluded from training or was previously unknown. In order to demonstrate the effect of missing classes in the training set in an authentic biological context, we analyzed a dataset containing seven morphological attributes of seeds from three types of wheat (Figs 2A and S1). In the original dataset from Charytanowicz *et al*. [50], three varieties of wheat—Kama (wheat 1), Rosa (wheat 2) and Canadian (wheat 3)—were grown and visualized using a soft X-ray technique, measuring area, perimeter, compactness, length of kernel, width of kernel, asymmetry coefficient and length of kernel groove. We divided the dataset into training and testing sets, then created three models, each trained on two types of wheat, and tested on all three. We then investigated the performance of SWIF(r) and the SRS in the case where one of the three classes is unrepresented in training, but present in testing. Fig 2B and 2C show that for each model, the instances belonging to the class that was excluded from training have significantly lower SRS values than instances that belong to a known class (p-value < 1e-08, Mann-Whitney Test; S1 Table). This is even true when the unknown class contains values that are intermediate to the two known classes (Fig 2B, leftmost panel), suggesting that the SRS is able to indicate deficiencies in the training set without relying on the presence of extreme attribute values. When SWIF(r) is trained on all three classes, no class has significantly different SRS values from the other classes (p-value > 0.05, Mann-Whitney Test; S1 Table and S2 Fig). We note the roughly bimodal shape of the SRS scores when a class missing from training data is represented by many instances in testing data (S3–S5 Figs). Combining this bimodal profile with domain expertise to inspect the low-SRS samples and identify their common features could lead to the identification of a novel class.

The "missing class" task is analogous to outlier or novelty detection, which can be evaluated by other methods including Local Outlier Factor [21], Isolation Forest [51,52], Elliptic Envelope [53] and Gaussian Mixture Models [54]. We find that the SRS has comparable performance to these methods on the task of distinguishing instances that came from a class that was not included in training (S3–S5 Figs), while having the advantage of being produced automatically within the SWIF(r) pipeline with the same underlying statistical framework. One additional advantage of the SRS over alternative methods, which we explore later, is that it can be applied even when instances contain missing values for some attributes; none of the other methods used in this comparison allow for any missing attribute values.

## The SRS is sensitive to systemic bias in training data relative to application data

In Fig 3, we demonstrate how the SRS can be used to identify systemic differences between training data and application data, using examples of demographic mismatch between training and testing sets in the UK Biobank. Two models were trained to predict elevated or normal blood sugar levels (HBA1C>48 mmol/mol for "Elevated", HBA1C<42 mmol/mol for "Normal") from a collection of phenotypes as attributes. Principal Components Analysis (PCA) for the phenotype data used in these experiments is available in S6 and S7 Figs. In the first "sex mismatch" model (Figs 3A, left and S6), the model was trained on data from males with European ancestry, using 22 attributes (see Methods) to predict Elevated or Normal blood sugar. This model was then tested on a second cohort of males with European ancestry, and a "non-matching" cohort consisting of females with European ancestry. In the second "ancestry mismatch" model (Figs 3A, right and S7), the model was trained on data from male and female individuals with European ancestry, using the same 22 attributes, with the same classification goal. This model was then tested on a second cohort of male and female individuals with European ancestry, and a non-matching cohort consisting of male and female individuals with African Ancestry (see Methods). In both cases, the average SRS of the non-matching cohort was significantly lower than the average SRS of the matching cohort (Fig 3A, p-values: S2 Table). This corresponded with systematic biases observed in SWIF(r)'s classification of the non-matching cohorts (Fig 3B), which resulted in an overabundance of individuals classified as Normal in the African ancestry cohort, and an overabundance of individuals classified as Elevated in the female sex cohort (Fig 3B). In order to investigate the causes of the reduction in SRS, we identified some attributes with large mean differences between matching and non-matching cohorts. In Fig 3C left, we show a pronounced example of mean difference between the distributions of cholesterol and hemoglobin across male and female cohorts that results in lower SRS values for the female cohort. In the background of the upper plots, we plot the value of the SRS across a plane defined by those two attributes, and overlay the distribution of trait values for each cohort. In the lower panels of Fig 3C, the background is colored using SWIF(r) probability. This allows us to visualize how the attribute values occupy different ranges of values of the SRS and of SWIF(r) probability. From this view, we can notice some patterns emerging in this mean shift example. The matching (male) cohort sits in the center of the area with highest SRS. Likewise, the probability distributions determined using the (male-only) training data appear to effectively separate the testing data from the male cohort. In contrast, the female cohort is shifted downward, out of the area of the plane with the highest SRS. This also causes the probability distribution shift relative to the means of this cohort, resulting in a greater number of individuals from both Elevated and Normal classes being classified as having elevated blood sugar levels. In the ancestry mismatch example, we observe that there is a smaller mean difference between the classes of the ancestry mismatch cohort (African/elevated vs.

African/normal) than there is between the matching ancestry cohort (European/elevated vs European/normal). This demonstrates a major failure for the classifier: by relying on a difference that is present in the training cohort, the classifier will not be able to accurately distinguish between individuals belonging to a cohort that lacks this difference. This results in the increased classification of individuals with African ancestry as Normal, for individuals in both the Normal and Elevated categories. We can see this in the lower right panel, where the distribution of individuals with elevated blood sugar levels is shifted towards the area given a high Normal probability (blue) or intermediate probability (white), rather than the area with high Elevated probability (red). The SRS lacks sensitivity in this scenario: *if differences between cohorts cause individuals from one class to look like valid instances from another class, SRS cannot help to identify these misclassified individuals.* PCA analysis (S6 and S7 Figs) provides an additional context for the challenge of classification and use of the SRS in the ancestry mismatch model: while there is a clear systemic difference between the Male and Female cohorts in attribute space (S6 Fig), the differences between African and European cohorts are not as obvious (S7 Fig). The lack of obvious differences between the African and European cohorts, and the fact that distribution shifts in this context cause individuals from one class to look like individuals from another class are both challenging situations where we might expect methods like the SRS that detect distribution shifts to fail.

When SWIF(r) is trained with data from both the matching and non-matching cohorts, the differences in the SRS are no longer significant for the experiment looking at sex (S8 Fig). The difference between the SRS for African and European cohorts is still significant for individuals in the Normal class, though not for individuals in the Elevated class (S9 Fig). We do not necessarily see an improvement in overall performance of the classifier with an increase in the diversity of the samples (S10 and S11 Figs). In fact, the classifier trained on both African and European ancestries performed worse on the European cohort, and only slightly better on the African cohort than the model trained on just the European cohort. This suggests that meaningful differences between the cohorts may be driving misclassification, possibly through a shift in which phenotypic measurements provide the best indicators of elevated HBA1C in a given population.

This experiment represents an important use case for the SRS; in many contexts, researchers may be unsure how well their classifiers will generalize to a new dataset. The SRS provides a quick way to view the overall difference in model fit between a training or validation set and a testing set, making it straightforward to test for significant shifts in SRS distribution using two-sample tests [17]. In this example, we are able to observe a statistically significant decrease in SRS score on cohorts mismatched on either sex or ancestry. This analysis is crucial for making subsequent decisions about the need for additional training.

## The SRS can be used to filter instances missing valuable attributes

We have shown how the SRS can assist researchers in detecting unknown classes in testing data (Fig 2), and identifying systemic mismatches between training and testing data (Fig 3). Another feature of the SRS that is especially relevant to real-world data is how it responds to data 'missingness'. Many machine learning classifiers and outlier/novelty detection methods do not allow for any attributes to be missing from an instance for classification to be performed. SWIF(r) allows for any number of missing attributes at a given instance, as long as at least a single attribute is defined, and it allows for the number of missing attributes to vary from instance to instance. However, while SWIF(r) is tolerant of missing data, performance generally declines as missingness increases [22]. Knowing this, it may be tempting to set a simple threshold on the number of missing attributes allowed, or otherwise filter instances that contain missing attributes. However, we find that the SRS offers a method for selectively

filtering instances that are missing valuable attributes, without losing instances that can be classified confidently with existing attributes.

In Fig 4, we illustrate this in a population genetics context, in which SWIF(r) is trained on simulated genetic data undergoing neutral evolution and balancing selection, with application data coming from the 1000 Genomes project [55] (Materials and Methods). In this dataset, each instance is a site in the genome. Missing data is ubiquitous in both simulated and application data due to the constraints on calculating many of the attributes, which in this case are selection statistics. In Fig 4A we observe the general problem that low SRS values are present, especially in application data. As we know from prior experiments, this can have multiple causes related to mismatch between training and testing data. In this case, application data had the highest average number of missing attributes, missing an average of 2.87 attributes out of five total attributes. Balancing selection simulations were missing an average of 2.75 attributes, and neutral evolution simulations were missing an average of 1.56 attributes. The number of missing attributes differed across data types and SRS ranges. Notably, out of 23 application data instances with SRS values less than -16, all 23 were missing selection statistics iHS and nSL, both measures of haplotype homozygosity.

In Fig 4B, we set up an experiment that demonstrates the benefit of using the SRS to determine how to filter instances with missing data, rather than setting an arbitrary threshold on missingness. First, we created a filtered training dataset from simulated data, using only instances for which all of the selection statistics are defined. We label our original set of selection statistics "Informative attributes". We also added 3 additional attributes that have no connection to instance properties; instead, they are drawn randomly from Gaussian distributions. These "Noise attributes" are drawn in an identical manner for both classes, so they have no information value to the machine learning algorithm. We filtered testing data in the same manner, with the same added Noise attributes. As a final step in the creation of testing data for this experiment, we added missingness by eliminating either three Informative attributes (Fig 4B, yellow and blue) or the three Noise attributes (Fig 4B, orange and red). Holding the number of missing attributes constant, we see that SRS is lower when Informative attributes are missing, and higher when Noise attributes are missing. A strict threshold on missingness would treat Noise and Informative attributes equally when deciding to eliminate an instance from consideration by the classifier. Instead, we demonstrate that we can use SRS to identify valuable attributes, in order to avoid eliminating instances that are missing low-value or uninformative attributes.

## Discussion

In this study we introduce the SWIF(r) reliability score (SRS), which assesses the trustworthiness of the classification for a specific instance using the model underlying the SWIF(r) classifier. We demonstrate the utility of the SRS when faced with common challenges in machine learning including: 1) an unknown class present in testing data that was not present in training data, 2) systemic mismatch between training and testing data, and 3) instances of testing data that have missing values for some attributes. Reliability scores in general provide a framework to evaluate how closely a given testing instance matches the training data underlying a classifier. The reliability score shares some features with outlier detection, novelty detection and detecting out-of-distribution instances [16,19–21,51–53], as well as methods for testing distribution shifts [17]. Our implementation, the SWIF(r) Reliability Score or SRS, is able to operate in the presence of missing attribute values, a strength that sets it apart from many related methods.

Current debates in the machine learning field propose two major distinct objectives for developing machine learning frameworks: the development of interpretable methods, which

have inherent features that allow users to understand how model conclusions are reached [4,24], and the development of explainable methods (and/or explanatory tools) which provide post-hoc or mathematically distinct overlays onto model conclusions in order to aid human understanding of how model conclusions may have been reached [56]. Most existing methods for the detection of outliers and novel instances in datasets operate in the second mode, providing a secondary analysis that creates a new model of the data, and then filters the data based on this model [16,21,51–53]. Methods for detecting out-of-distribution instances in the context of neural networks [18–20] bear some resemblance to the SRS, as they attempt to use aspects of the neural network architecture to derive measures of confidence [19] or to identify out-of-distribution instances [18]. In this paper, we contribute towards the goal of interpretability specifically: because of SWIF(r)'s generative framework, we have an available model upon which to build the SRS, which can be used to scrutinize multiple failure cases of the SWIF(r) machine learning algorithm. The SRS allows a user to probe for important deficiencies in the training and testing data, including missing classes, systemic mismatch between training and testing data, and instances missing informative attributes. For example, users may look for a bimodal distribution in SRS values to indicate a large number of instances that may come from an unknown class, allowing for subsequent investigation of what those instances have in common. Additionally, analyzing SRS distributions between known subgroups may help users diagnose a systemic mismatch between training and application data. While a full interpretation of the causes contributing to a low SRS requires domain specific knowledge, use of the SRS offers researchers several potential remedies, including removal of outliers, measurement of distribution shifts, expanding or revising the training set, or the creation of multiple classifiers to account for distinct subgroups in their analysis.

The SRS represents a valuable measure for SWIF(r), and similar reliability scores could be created for closely related methods with a generative basis. This underscores the inherent interpretability of generative methods: because they learn the underlying distribution of data for each class, reliability scores can reveal elements of that process in order to improve the quality of classification, and understand its basis. Since discriminative classifiers learn decision boundaries directly without learning underlying distributions, these methods (including most neural networks, support vector machines, random forest classifiers, regression-based methods and more) require a different approach, such as post-hoc use of novelty detection methods and measures of distribution shifts. In such a case, it is possible to use the SRS independent of applying SWIF(r), as a complementary read-out alongside another machine learning classifier. In this case, the SRS would be an explanatory tool for examining results or refining the design of the training set, but could not be understood as providing insight into the specific classifier. We note that the SRS, similar to SWIF(r) itself, is particularly suitable as an explanatory tool for applications with a small-to-moderate number of features. With a large number of features, pairwise correlations learned by the SWIF(r) framework may not be sufficient to capture the full structure of the data, and other tools for outlier detection or distribution shift might be more desirable. On the other hand, in situations in which the features are highly correlated, it might be preferable to reduce the number of features to a smaller, less correlated set using an approach such as PCA.

One limitation of the SRS is that while the SRS indicates problems with missing data or deficiencies in the dataset, a solution to those problems may not always be possible or simple to implement. Researchers will need to rely on domain-specific knowledge to decide whether the indicated problem can be solved by removing instances, adding additional classes or attributes to their model, or by including more, or more diverse instances, in their training data. It may also be advantageous to train independent classifiers for cohorts that may have distinct

underlying properties; for example, separate classifiers for each sex may be useful in a study where the goal is to distinguish individuals with a disease phenotype [57] (Fig 3). Discovering a solution to low-reliability instances will require answers to subjective and context-dependent questions that may be difficult to anticipate and may not generalize. Care should be taken to use abstention, or the removal of instances in a principled and appropriate manner. When there is risk involved in making the wrong classification, abstention can be an important practice, as in medical contexts [46,47]. In other cases there may be a distinctive factor impacting the performance of the classifier on specific instances: for example, when simulated training data fails to capture dynamics in unusual genomic regions. We note that abstention or removal of instances should not be used to boost measurement of performance of the classifier, but to invite further investigation and scrutiny. When distribution shifts or additional classes are suspected, researchers may face decisions about the number and types of classes to include in training, and cost-benefit considerations when deciding whether to gather the additional data required to add new attributes or additional training examples. Increasing transparency and publication of models with negative results would help address some of the problems raised by these subjective questions.

Our study is motivated by the fact that the investigation of flaws in machine learning has become an important topic of research in its own right. As machine learning methods are put to the test for applications like medical image analysis, we encounter examples where the accuracy and value of machine learning methods are artificially inflated by flaws in the training set, overconfidence in results, or reliance on unintended features of data [58,59]. Many of these machine learning methods are so-called 'black box' methods that are relatively difficult to deconstruct. Explanatory tools built to analyze the output of these methods often rely on post-hoc challenges to a trained model. In contrast, the relative mathematical simplicity of SWIF(r) and the SRS allows for more direct observation of the performance of the machine learning classifier.

Across many areas of biology there is an opportunity to improve machine learning analyses by creating community standards and best practices for simulation, interpretability and reproducibility. By packaging SWIF(r) and the SRS together into a single workflow, we aim to give researchers the tools to both apply the machine learning method, and to critique and interpret the results they obtain. In addition, by including the SRS as default output when using the SWIF(r) classifier, we hope to increase the adoption of this way of thinking, encouraging researchers across the biological spectrum who use out-of-the-box classifiers to interrogate results that might be too good to be true, or understand the ways in which their training data may be systematically biased. As machine learning applications to biological data expand, it will be important to prioritize the ability to draw biological conclusions from data. We also cannot afford to lose sight of the way that subjective researcher-specific decisions guide the design of machine learning models. Domain-specific biological knowledge currently provides the best available guide for navigating the complex, multi-factorial choices involved in model design. In order to best leverage this domain-specific expertise, we need to ensure that machine learning models are easy to interpret, dissect and critique, including by researchers without machine learning expertise. Tools that enhance the interpretability and transparency of machine learning algorithms are an important step towards this goal.

## Materials and methods

### Overview and Definition of SRS

SWIF(r) was originally defined to calculate the probability that a genomic site is adaptive (vs neutral) using the following formula, with the terms adjusted here to suit the more general

case and to match the definition of the SRS in Eq 2:

$$P(class\ 1|X_1 = x_1, \ldots, X_n = x_n)$$

$$= \frac{\pi \sum_{j=1}^{n}[f(X_j = x_j|class\ 1)\prod_{i \neq j}f(X_i = x_i|X_j = x_j, class\ 1)]}{\pi \sum_{j=1}^{n}[f(X_j = x_j|class1)\prod_{i \neq j}f(X_i = x_i|X_j = x_j, class1)]} \tag{Eq3}$$

$$+ (1-\pi)\sum_{j=1}^{n}[f(X_j = x_j|class\ 2)\prod_{i \neq j}f(X_i = x_i|X_j = x_j, class\ 2)]$$

This framework can be generalized to other applications by simply changing the classification categories (in [22], adaptive and neutral), and with adjustment of the Bayesian prior $\pi$ according to the specific application. Attributes denoted by X must be continuous variables. Eq 1 (Results) adapts the numerator of SWIF(r), which approximates the likelihood of the observed data conditioned on each class using both the marginal distributions of the attributes and the two-dimensional joint distributions of the attributes, from which conditional distributions are derived. The SWIF(r) probability is the posterior probability of a class given the set of attributes. In the case of an instance with one or more missing attributes, both SWIF(r) and the SRS are calculated based on the set of present attributes rather than the full set of n attributes, altering the number of density functions in each sum and product of Eqs 2 and 3. The output from Eq 3 can be thought of as answering the question: Given the training data and the available classes, what is the probability that an instance came from this class (instead of another of the available classes)? Eq 2 (Results), which defines the SRS, aims to ask a different question, which is simply: What is the likelihood of seeing an instance like this one, across all available classes?

In Fig 1 of this study, SWIF(r) was trained using a $\pi$ value of 0.5, an unbiased prior that gives both classes equal weight. In Fig 1A, data for each class was generated with a single attribute, drawn from N(-1, $0.2^2$) for class 1 and N(+1, $0.2^2$) for class 2. In Fig 1C, we generated two attributes for each class instance from bivariate Gaussian distributions. In this experiment, three different models were trained, each consisting of two classes (S3 Table). 1000 instances were generated for each class to train SWIF(r). To create the testing set, 2000 instances were generated by drawing from the uniform distribution across the plane bounded by (-2, 2) in both dimensions. SWIF(r) classification and the SRS were both calculated on each instance in the testing set, using the x and y coordinates as the attribute values. Data and code for this analysis are available on Github: https://github.com/ramachandran-lab/SRS_paper. SWIF(r), with the SRS included, can be installed according to instructions available at: https://github.com/ramachandran-lab/SWIFr.

### Unknown class example: Wheat seeds dataset

Data for this experiment was originally gathered by Charytanowicz et al. [50]. Three varieties of wheat: Kama (wheat 1), Rosa (wheat 2) and Canadian (wheat 3), were grown in experimental fields at the Institute of Agrophysics of the Polish Academy of Sciences in Lublin. 70 seeds from each variety were randomly selected, and visualized using a soft X-ray technique. Seven geometric parameters were measured: area, perimeter, compactness, length of kernel, width of kernel, asymmetry coefficient and length of kernel groove [50].

To construct training and testing sets, each set of 70 was randomly divided into two sets of 35, for training and testing respectively. We created three models, each trained on two types of wheat. Testing was then performed on all three types. See S4 Table for sample breakdown. Differences in class distributions of SRS used Mann-Whitney U tests. For performance comparison the following methods were used: Elliptic Envelope, Local Outlier Factor and Isolation

Factor from sklearn [60] and Gaussian Mixture Models from sklego (scikit-lego: https://scikit-lego.netlify.app). Data and code for this analysis are available on Github: https://github.com/ramachandran-lab/SRS_paper.

## Systemic mismatch between training and testing set: UK biobank elevated blood sugar

Phenotype data was downloaded from the UK Biobank under Application 22419, for 22 phenotypes used as attributes, and HBA1C as an outcome. Attributes were as follows: BMI (Body Mass Index), MCV (Mean Corpuscular Volume), Platelet (count of Platelets in blood), DBP (Diastolic Blood Pressure), SBP (Systolic Blood Pressure), WBC (White Blood Cell count), RBC (Red Blood Cell count), Hemoglobin (Hemoglobin blood test), Hematocrit (hematocrit blood test), MCH (Mean Corpuscular Hemoglobin), MCHC (Mean Corpuscular Hemoglobin Concentration), Lymphocyte (Lymphocyte cell count), Monocyte (Monocyte cell count), Neutrophil (Neutrophil cell count), Eosinophil (Eosinophil cell count), Basophil (Basophil cell count), Urate (Uric acid concentration in blood), EGFR (Estimated Glomerular Filtration Rate), CRP (C-Reactive Protein concentration), Triglyceride (Triglyceride concentration), HDL (High-Density Lipoprotein test), LDL (Low-Density Lipoprotein test) and Cholesterol (Cholesterol test). For each phenotype, we centered the mean phenotype value at 0 and scaled the phenotype values to have a standard deviation of 1. We then selected a subset of those individuals for training each of two models. Within each model, individuals were divided on the basis of their HBA1C measurement, a blood test that estimates blood sugar levels. Individuals with an HBA1C reading of over 48 mmol/mol were sorted into the "elevated" cohort; readings of 48 mmol/mol or greater are typically considered diagnostic for diabetes. Individuals with an HBA1C reading of 42 mmol/mol or below were sorted into the "normal" cohort. Individuals with HBA1C readings of 42–48 mmol/mol, a range associated with prediabetes, were not included in the analysis. See S5 Table for sample breakdown. All statistical tests performed in this section are two-sample t-tests.

**Ancestry mismatch experiment.** Individuals were first divided on the basis of ancestry, as in [61], resulting in 349,411 individuals of self-identified European descent, and 4,967 individuals of African descent. The latter group was identified both by self-identification and by an ADMIXTURE analysis as described in [61]. Applying the HBA1C filter described above resulted in 8,631 individuals in the European/elevated cohort, 268 individuals in the African/elevated cohort, 243,283 individuals in the European/normal cohort and 2,532 individuals in the African/normal cohort. For the purposes of training, 800 individuals were selected at random from each of the European/elevated and European/normal cohorts, creating a training set with 1600 European individuals. The model was then tested on 800 individuals from each cohort, with the exception of the African/elevated cohort which was tested on the 268 eligible individuals.

**Sex mismatch experiment.** The same 349,411 individuals of European descent described in the ancestry experiment were divided on the basis of sex, yielding 134,578 female individuals and 117,336 male individuals. Applying the HBA1C filter described above resulted in 131,432 individuals in the female/normal cohort, 3146 individuals in the female/elevated cohort, 111,851 individuals in the male/normal cohort and 5485 individuals in the male/elevated cohort. For the purposes of training, 800 individuals were selected at random from each of the male/normal and male/elevated cohorts, creating a training set with 1600 male individuals. The model was then tested on 800 individuals from each cohort.

## Missing data: Balancing selection/1000 genomes dataset

To produce genome simulations of balancing selection and neutral evolution we used SLiM, pyslim and msprime to combine coalescent and forward simulation approaches [62]. We

simulated 1 Mb genomic regions with 500 simulations for each of the two conditions (balancing and neutral). Balancing simulations contained a single balanced mutation at the center of the 1 Mb genomic region. The mutation was introduced 100,000 generations ago, with a selection coefficient (s) of 0.01, and overdominance coefficient (h) varied according to a uniform distribution from 1.33 to 5. The simulation process used SLiM 3.3, including tree sequence recording, along with pyslim, tskit and msprime for recapitation and neutral mutation overlay [62]. We simulated African, European, and Asian populations according to the demographic model developed by Gravel *et al.* [63]. Following simulation, we sampled 200 individuals (400 haplotypes) from the African population for each selection scenario to generate training data. Testing data was generated in an identical manner, with 100 simulations performed for each selection scenario.

Cross-Population Extended Haplotype Homozygosity (XP-EHH) [28], Integrated Haplotype Score (iHS) [26], iHH12 [64,65] and nSL [66] were calculated using selscan [67]. These statistics require phased genomic data as input, and were designed to detect signatures of recent or ongoing positive selection in genomes. Beta, a statistic built to detect balancing selection using ancestral/derived allele frequencies, was calculated using Betascan [68]. For XP-EHH, which requires an outgroup population to calculate, the simulated European population served as the outgroup for the African population sampled. DDAF (difference in derived allele frequency) also requires an outgroup allele frequency to calculate. Here, the average of the European and Asian allele frequencies was used.

For comparison with 1000 Genomes data, all of the above summary statistics were calculated on the complete set of YRI individuals. Where outgroup calculations were required, the CEU and CHB populations were used in the same manner as the European and Asian simulations, respectively. Simulated testing data for each scenario, as well as 1000 Genomes data, were processed with SWIF(r), producing classification probabilities and SRS values (shown in Fig 4A). Data and code for this analysis are available on Github: https://github.com/ramachandran-lab/SRS_paper.

## Supporting information

**S1 Table. Mann-Whitney test scores comparing SRS distributions for wheat classes under different models.** Classifiers were trained with two out of three classes and tested on all three classes. The missing class is indicated in the first column. Highlighted cells indicate comparisons where we do not expect significant differences between classes because they are both present in the training data.
(PDF)

**S2 Table. Average SRS values and p-values for UKB cohort comparisons.**
(PDF)

**S3 Table. Gaussian distributions by class and model for Fig 1C.**
(PDF)

**S4 Table. Sample sizes for wheat dataset.**
(PDF)

**S5 Table. Sample sizes for UKB cohorts.**
(PDF)

**S1 Fig. View of wheat dataset.** Scatterplots of each pair of attributes. Histograms showing the distribution for each individual attribute.
(PDF)

**S2 Fig. SRS scores show a unimodal pattern when trained on all three classes of wheat.** Histogram of SRS scores generated when SWIF(r) was trained on all three classes of wheat, and tested on all three classes.
(PDF)

**S3 Fig. SRS has similar performance to other outlier detection methods when separating instances from an unknown class (wheat 1).** (Top) Receiver Operating Characteristic curve of outlier detection methods and the SRS. The "True Positive Rate" is the fraction of members of the trained classes encompassed by the score threshold, while the "False Positive Rate" is the fraction of the members of the "unknown" or excluded class encompassed by the score threshold. (Middle) The unknown class is shown in blue (wheat 1), while the classes included in training data are shown in red and green (wheat 2, wheat 3). We see that for all methods, the majority of instances of wheat 1 (blue) are to the left, receiving lower scores than instances from known classes. (Bottom) Receiver Operating Characteristic curve of Gaussian Mixture Models (GMM) outlier detection methods and the SRS. The "True Positive Rate" is the fraction of members of the trained classes encompassed by the score threshold, while the "False Positive Rate" is the fraction of the members of the "unknown" or excluded class encompassed by the score threshold. GMM10 refers to a Gaussian Mixture Model trained with 10 components, GMM5 is trained with 5 components and GMM20 is trained with 20 components.
(PDF)

**S4 Fig. SRS has similar performance to other outlier detection methods when separating instances from an unknown class (wheat 2).** (Top) Receiver Operating Characteristic curve of outlier detection methods and the SRS. The "True Positive Rate" is the fraction of members of the trained classes encompassed by the score threshold, while the "False Positive Rate" is the fraction of the members of the "unknown" or excluded class encompassed by the score threshold. (Middle) The unknown class is shown in red (wheat 2), while the classes included in training data are shown in blue and green (wheat 1, wheat 3). We see that for all methods, the majority of instances of wheat 2 (red) are to the left, receiving lower scores than instances from known classes. (Bottom) Receiver Operating Characteristic curve of Gaussian Mixture Models (GMM) outlier detection methods and the SRS. The "True Positive Rate" is the fraction of members of the trained classes encompassed by the score threshold, while the "False Positive Rate" is the fraction of the members of the "unknown" or excluded class encompassed by the score threshold. GMM10 refers to a Gaussian Mixture Model trained with 10 components, GMM5 is trained with 5 components and GMM20 is trained with 20 components.
(PDF)

**S5 Fig. SRS has similar performance to other outlier detection methods when separating instances from an unknown class (wheat 3).** (Top) Receiver Operating Characteristic curve of outlier detection methods and the SRS. The "True Positive Rate" is the fraction of members of the trained classes encompassed by the score threshold, while the "False Positive Rate" is the fraction of the members of the "unknown" or excluded class encompassed by the score threshold. (Middle) The unknown class is shown in green (wheat 3), while the classes included in training data are shown in blue and red (wheat 1, wheat 2). We see that for all methods, the majority of instances of wheat 3 (green) are to the left, receiving lower scores than instances from known classes. (Bottom) Receiver Operating Characteristic curve of Gaussian Mixture Models (GMM) outlier detection methods and the SRS. The "True Positive Rate" is the fraction of members of the trained classes encompassed by the score threshold, while the "False Positive Rate" is the fraction of the members of the "unknown" or excluded class encompassed

by the score threshold. GMM10 refers to a Gaussian Mixture Model trained with 10 components, GMM5 is trained with 5 components and GMM20 is trained with 20 components.
(PDF)

**S6 Fig. Principle Components Analysis of dataset composed of health attributes for male and female individuals of European descent with both normal and elevated HBA1C.** For each class, 800 individuals meeting the class definition (sex and HBA1C status) were selected at random from the set of individuals who self identified. PCs 1–10 are shown.
(PDF)

**S7 Fig. Principle Components Analysis of dataset composed of health attributes for individuals of both African and European descent with both normal and elevated HBA1C.** For each class, 800 individuals meeting the class definition (ancestry and HBA1C status) were selected at random. Only 268 individuals met the class definition of African ancestry and elevated HBA1C, so only these 268 individuals were included for that class. PCs 1–10 are shown. Individuals of both sexes were included in the analysis.
(PDF)

**S8 Fig. SWIF(r) model trained with both male and female European individuals and tested on both male and female individuals.** Boxplots (left) and histograms (right) represent the same data, with histograms zoomed out to show outliers, while outliers are not shown in boxplots. Annotation "ns" indicates "not significant" (t-test: top: p = 0.353, bottom p = 0.254).
(PDF)

**S9 Fig. SWIF(r) model trained with both African and European individuals and tested on both African and European individuals.** Boxplots (left) and histograms (right) represent the same data, with histograms zoomed out to show outliers, while outliers are not shown in boxplots. Annotation "ns" indicates "not significant," "*" indicates a significant difference (t-test: top: p = 0.165, bottom p = 0.00216*)
(PDF)

**S10 Fig. Confusion matrix for African and European test sets with SWIF(r) model trained with both African and European samples.** A SWIF(r) model was trained with combined African and European data, and tested on the African and European test cohorts from Fig 3. Numbers correspond to the following: 0 = Elevated HBA1C, 1 = Normal HBA1C. Actual class is on the y-axis, predicted class is on the x-axis.
(PDF)

**S11 Fig. Confusion matrix for Male and Female European test sets with SWIF(r) model trained with both Male and Female European samples.** A SWIF(r) model was trained with combined Male and Female European data, and tested on the Male and Female European test cohorts from Fig 3. Numbers correspond to the following: 0 = Elevated HBA1C, 1 = Normal HBA1C. Actual class is on the y-axis, predicted class is on the x-axis.
(PDF)

## Acknowledgments

The authors thank Samuel Pattillo Smith for assistance with UKBiobank data.

## Author Contributions

**Conceptualization:** K. D. Ahlquist, Lauren A. Sugden, Sohini Ramachandran.

**Formal analysis:** K. D. Ahlquist, Lauren A. Sugden.

**Funding acquisition:** Sohini Ramachandran.

**Investigation:** K. D. Ahlquist, Lauren A. Sugden.

**Methodology:** Lauren A. Sugden.

**Resources:** Sohini Ramachandran.

**Software:** K. D. Ahlquist, Lauren A. Sugden.

**Supervision:** Sohini Ramachandran.

**Validation:** K. D. Ahlquist.

**Visualization:** K. D. Ahlquist, Lauren A. Sugden.

**Writing – original draft:** K. D. Ahlquist, Lauren A. Sugden.

**Writing – review & editing:** K. D. Ahlquist, Lauren A. Sugden, Sohini Ramachandran.

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
