## [Decision Letter · Decision Letter 0]

16 Jun 2022

Dear Dr. Sugden,

Thank you very much for submitting your manuscript "Enabling interpretable machine learning for biological data with reliability scores" for consideration at PLOS Computational Biology. We apologize for the delay in handling your manuscript.

As with all papers reviewed by the journal, your manuscript was reviewed by members of the editorial board and by several independent reviewers. In light of the reviews (below this email), we would like to invite the resubmission of a significantly-revised version that takes into account the reviewers' comments.

We ask that the authors address all the concerns raised by the reviewer, but note that a common theme is a lack of comparison to existing methods and approaches.

We cannot make any decision about publication until we have seen the revised manuscript and your response to the reviewers' comments. Your revised manuscript is also likely to be sent to reviewers for further evaluation.

Sincerely,

Luis Pedro Coelho

Associate Editor

PLOS Computational Biology

Sushmita Roy

Deputy Editor

PLOS Computational Biology

We ask that the authors address all the concerns raised by the reviewer, but note that a common theme is a lack of comparison to existing methods and approaches.

Reviewer's Responses to Questions

**Comments to the Authors:**

Reviewer #1: Title: Enabling interpretable machine learning for biological data with reliability scores

Summary

The manuscript proposes a reliability score, SWIF(r) reliability score (SRS), to assess a Machine learning classifier’s ability to produce a reliable classification for test data. SRS builds on a previous generative model SWIF(r) from the same group which produces a posterior probability for each class in the classification task. SRS in comparison focuses only on the estimated likelihoods. The utility of SRA is shown using three common scenarios from machine learning: 1) presence of a new class in test data absent from the training data, 2) distributional mismatch between training and test data, and 3) missing attributes (features) in test data. The first scenario is tested using a classification task to distinguish between three wheat seed varieties using seed morphology features. Training data consists of two seed types and test data contains all three. SRS assigns low reliability to predictions of the wheat seed type missing from the training data whereas SWIF(r) probability confidently assigns them to the other two classes for the majority of test instances. The second scenario is showcased with UK Biobank data, where the train and test data have a demographic mismatch (training on male individuals of European ancestry for elevated versus normal sugar and testing on either sex-mismatched or ancestry mismatched data). Instances from matched data (European male) result in a higher reliability score in comparison to mismatched (sex and ancestry) instances. The final scenario is illustrated by showing that SRS can deal with missingness in the test data and instances where informative features are removed have lower SRS than instances where noise features are removed.

Major comments:

1. The code repository is not available (https://github.com/ramachandran-lab/SRS_paper).

2. While the SRS score is appealing to estimate the reliability of a model’s prediction, it is limited in application because of its dependence on the SWIF(r) generative model. Can it be extended to be applicable to other model types?

3. The manuscript is lacking in any sort of comparison with existing methods for estimating uncertainty in a classifier’s prediction (For example, https://arxiv.org/pdf/1805.11783.pdf). Furthermore, there are several published papers about the calibrating for the distributional shift between train and test data (for example, https://proceedings.neurips.cc/paper/2019/file/846c260d715e5b854ffad5f70a516c88-Paper.pdf). Comparing against at least some of them will give us an idea of the utility of SRS.

4. The paper needs to expand the section explaining SWIF(r) and SRS, currently, the explanation is not self-sufficient and the reader needs to go back to the SWIF(r) manuscript to properly understand the contribution of the current work.

Minor comments:

1. The references to figure 1 in the text are incorrect. In the section, Overview and Definition of SRS, the text refers to:

a. Figure 1B instead of Figure 1A.

b. The two classes are labeled as class 1 and class 2 in Figure 1A but referred to as classes A and B in the text.

c. The second panel of Figure 1B shows SWIF(r) probability and the third panel shows the SRS score however, the text has the two references flipped.

2. Figure 2 panels are not probably explained in the text.

Reviewer #2: - This work presents a novel, generalizable metric “reliability score” (or “SRS” in this particular application), used to diagnose several types of issues in machine learning workflows, especially those due to differing contexts between training and testing data (missing classes, cohort effects, and missing attributes). Its successful application to three distinct datasets promises broad utility in supporting ML analyses. However, some claims regarding the SRS require additional control experiments to determine whether differences in SRS distributions are due to limitations in the training data or genuine class/cohort differences.

- SRS definition, Paragraph 4: In Model Y in Figure 1B, the area directly between the centers of the two classes cannot be classified (SWIF(r) probability ~ 0.5), yet the SRS for that area is relatively high. The results text focuses on cases with high SWIF(r) probability but low SRS, and requires an interpretation of this reversed example with low SWIF(r) probability but high SRS for a complete picture on how to interpret the possible combinations of prediction probabilities and SRSs.

- SRS and unknown classes, Paragraph 1: It’s not clear why the specific attribute pairs in Figure 2A and 2C were selected to represent the wheat data. If the aim is to show broadly how the data are distributed, alternate approaches include projecting the data onto the first two components from PCA to capture maximum variance, or generating all individual and pairwise attribute plots (i.e. Seaborn “pairplot”) as a supplementary figure. Please address or clarify.

- SRS and unknown classes, Paragraph 1: This section and Figure 2B-C suggests a strong separation in SRS between predictions for instances from classes known vs unknown to the training data. Please include statistical tests (i.e. Mann-Whitney) to quantify the significance of this separation in SRS values between known vs. unknown classes and known vs. known classes. Additionally, please include similar SRS plots for a SWIF(r) model trained with all three classes labeled as a control model and discuss whether the SRS values per class become less separable with full label information or if the difference persists (possibly due to genuine class-effects on SRS values).

- SRS and systemic bias, Paragraph 1: Similar to Figure 2A, include a figure showing systemic differences between the cohorts (Male vs Female, European vs African) in attribute space preceding Figure 3A (for example, see previous comment on PCA projections).

- SRS and systemic bias, Paragraph 1: This section and Figure 3A-B suggests that SRS distributions may be able to diagnose systemic bias when applying a model to data from a different cohort than the training set. Similar to before, please add corresponding figures for control models trained with data from both cohorts and discuss whether the mixed training data yields smaller differences in SRS values and/or misclassification rates between cohorts.

- SRS and missing attributes, Paragraph 3: This section and Figure 4B suggests that the SRS can distinguish informative from uninformative attributes. I would be interested in seeing a more fine-grained assessment of this capability in addition to the informative vs. pure noise test provided. For example, this could involve selecting all instances with 3/5 attributes missing, separating them into 10 groups based on which exact set of 3 attributes are missing, and then comparing the SRS distributions between those 10 groups, to test if SRS can distinguish how informative different subsets of attributes are beyond a simple missingness filter.

Minor comments:

*SRS definition, Paragraph 3: Classes are referred to as “class A” and “class B”, while Figure 1/Table S2 refers to them as “class 1” and “class 2”.

*SRS and unknown classes, Paragraph 1: Expand introduction of wheat dataset and include dataset size (number of instances per wheat class, in training/testing datasets)

*SRS and systemic bias, Paragraph 1: Include dataset size (number of Elevated and Normal instances for Male/Female and European/African cohorts), possibly as table that could be referenced both here and in the Methods.

*Methods, SRS, Paragraph 1: Equation 3 could be easier to understand if “adaptive”/”neutral” were replaced with “class 1”/”class 2” and the “S” terms replaced with “X” terms, to be more consistent with Equations 1/2 and Figure 1.

*Methods, SRS, Paragraph 2: Classes are referred to as “class A” and “class B”, while Figure 1/Table S2 refers to them as “class 1” and “class 2”.

*Figure 1B: Add axis labels, “attribute 1” and “attribute 2”

*Figure 2C: Add axis labels, at least to the True Labels plot.

*Figure 3A: X-axes four all four panels seem zoomed out, zooming in on peaks could better show the separation in SRS values.

*Figure 3C: Add axis labels.

*Figures 1-3 need larger axis tick labels.

Reviewer #3: In this work, the authors come up with a per-exemplar score called SRS by fitting a generative classifier (specifically an Averaged One-Dependence Estimation model) to labeled data. The SRS is then defined as the max log likelihood over classes. The authors demonstrate the general applicability of SRS as a tool to detect outliers and differences in training and application data.

By itself, the work is sound and the applications shown are reasonable and useful. However, my primary concern with this work is that it fails to discuss and/or compare against relevant literature in this vast field. The claim that SRS is a "new concept for scientific machine learning studies" lacks support. A particularly similar idea to SRS is using likelihood from mixture models to detect outliers (see https://scikit-lego.readthedocs.io/en/latest/mixture-methods.html#outlier-detection). SRS is essentially a simpler version of this where the class labels are known (uses max likelihood over classes instead of likelihood).

The work has close ties to multiple areas, none of which are discussed in the paper:

- outlier/novelty detection: Many methods available here https://scikit-learn.org/stable/modules/outlier_detection.html. Another related idea is one class SVM: https://www.microsoft.com/en-us/research/wp-content/uploads/2016/02/tr-99-87.pdf. Longer review https://www.robots.ox.ac.uk/~davidc/pubs/NDreview2014.pdf.

- testing differences between high dimensional distributions (aka two-sample test): See https://normaldeviate.wordpress.com/2012/07/14/modern-two-sample-tests/ for a helpful overview of methods such as MMD (https://www.jmlr.org/papers/volume13/gretton12a/gretton12a.pdf), cross-match test.

- abstention/robustness/out-of-distribution and domain shift/adaptation in ML: see e.g. https://arxiv.org/abs/2105.07107, https://arxiv.org/abs/1802.04865, https://arxiv.org/abs/1706.02690, https://arxiv.org/abs/1807.03888

Overall, it is not clear what the primary contribution of the proposed method is over the plethora of alternatives that serve a similar function. It would be helpful to contextualize this work and discuss the pros and cons over existing methods.

Reviewer #4: The authors discussed a metric, reliability score (SRS), to assess the reliability of classification of a given instance. Specifically, the authors point out three importance challenge in machine learning that SRS can help resolve: 1) an unknown class present in testing data that was not present in training data, 2) systemic mismatch between training and testing data, and 3) instances of testing data that are missing values for some attributes. I have three major comments that I hope that authors can provide more explanations on.

First, there is no question that, when we know the ground truth in either cases, SRS can help distinguish the presence of an unknown class, or indicate the presence of systemic bias. However, one of the most challenging issues is that frequently we cannot distinguish 1) and 2). When we have poorly classified test instances, we almost always do not know if it is a new class or the training data simply cannot lead to a more general model. My understanding is that SRS cannot help us in this regard. So, the claim that SRS can help resolve 1) and 2) only applies when we know a prior what the situation is, which make it not particularly useful.

Second, the SRS appears useful in identifying instances that do not resemble training data. However there are many other measures one can use, including very simple ones such as Euclidean distance or similarity measures like correlation, to achieve the same goal. SRS’ calculation requires priors and calculation can be involved. Is it better than the other approaches? If so, can there be some ways to formally demonstrate this?

Third, I would love to have some reliability measure for each instance in my classification tasks. But given the first point, while I appreciate the conceptual basis of a reliability score, SRS seems to be an outlier detection measure where there are already plenty.

More details:

1. p.2, “instances of testing data that are missing values for some attributes” – should it be “have missing values”?

2. p.5, “When these differences arise, it may be preferable to set aside data instances with poor fit,

allowing the classifier to effectively abstain when it encounters signatures it has not been trained

to expect, one goal of the SRS is to enable the user to identify such poor-fit instances”: This aspect needs to be more clearly explained and nuances listed. The practice can also make a model looks better than it actually is. It is not straightforward to distinguish true outliers from training data that do not sufficiently represent the true distribution.

3. p.7, equation 1: I am confused as to whether the sum term and the multiplication terms are done separately, or P(xi|class) is multiplied with all non-i attributes’ prior individuals, then summed together.

4. p.8, “Because the posterior probabilities are forced to sum to 1, instances that do not resemble any trained class may still receive a high classification probability for the “least bad” option…”: The authors then indicated that “SRS is designed to help detect such instances”. In Figure 1B (by the way, 1A would be out of order and has not been introduced yet), the authors nicely shown that some test cases that do not fall within the range of the training data would have poor SRS. So, an instance can have high classification probability but low SRS. The confusing thing here is then what is the “classification probability” the authors speak of? Some clarification here will be helpful to confused readers like myself.

5. p.8, another thought about Figure 1B: SRS is a measure that is useful for picking out which instances are similar to training data. That is fine, but we can imagine this can also be done using some simple similarity measures to training instances without the need to calculate posterior probabilities. If that is the case, what is special about SRS?

6. p.9 and on, the authors discussed the utility of SRS in picking out instances from unknown classes. There is no question that the SRS scores for the unknown classes are lower. But one distinction I’d like to make is that this does not mean that all instances with low SRS scores belong to some unknown classes (this seems to be what the authors are suggesting). Another possibility of low SRS scores can also be attributed to non-representative training data, which would make some test cases look like outliers when they actually are not. The challenge is to distinguish between these two possibilities. Can the authors provide some clarification as to how SRS can be helpful here?

7. p.10, the author indicated that, when there are systematic differences between training and application data, SRS can indicate them. Again, there is no question that SRS can show that there are systematic differences as Figure 3 shows when we know that is the case. The issue is however, when we do not know, how can one use SRS to distinguish the possibility between unknown class (point 6) and systematic bias.

**Have the authors made all data and (if applicable) computational code underlying the findings in their manuscript fully available?**

Reviewer #1: **No: **The code is not available at the provided link (https://github.com/ramachandran-lab/SRS_paper)

Reviewer #2: Yes

Reviewer #3: Yes

Reviewer #4: Yes

PLOS authors have the option to publish the peer review history of their article (what does this mean?). If published, this will include your full peer review and any attached files.

Reviewer #1: No

Reviewer #2: No

Reviewer #3: No

Reviewer #4: No
---

## [Decision Letter · Decision Letter 1]

7 Feb 2023

Dear Dr. Sugden,

Thank you very much for submitting your manuscript "Enabling interpretable machine learning for biological data with reliability scores" for consideration at PLOS Computational Biology.

As with all papers reviewed by the journal, your manuscript was reviewed by members of the editorial board and by several independent reviewers. In light of the reviews (below this email), we would like to invite the resubmission of a significantly-revised version that takes into account the reviewers' comments.

The reviewers generally agree that the revision has improved the manuscript and we agree with their assessment. We also feel that, generally speaking, the manuscript properly acknowledges the limitations of the method. Thus, while we can see the merit in the concerns of reviewer #4, we nonetheless consider that, on balance, this paper makes a sufficient contribution to the literature that it will be of interest to many readers if the remaining issues with presentation can be fixed. The authors may decide to add new data and analyses to bolster their arguments, but we consider that the remaining issues can likely be addressed with textual improvements.

We cannot make any decision about publication until we have seen the revised manuscript and your response to the reviewers' comments. Your revised manuscript is also likely to be sent to reviewers for further evaluation.

Sincerely,

Luis Pedro Coelho

Academic Editor

PLOS Computational Biology

Sushmita Roy

Section Editor

PLOS Computational Biology

The reviewers generally agree that the revision has improved the manuscript and we agree with their assessment. We also feel that, generally speaking, the manuscript properly acknowledges the limitations of the method. Thus, while we can see the merit in the concerns of reviewer #4, we nonetheless consider that, on balance, this paper makes a sufficient contribution to the literature that it will be of interest to many readers if the remaining issues with presentation can be fixed. The authors may decide to add new data and analyses to bolster their arguments, but we consider that the remaining issues can likely be addressed with textual improvements.

Reviewer's Responses to Questions

**Comments to the Authors:**

Reviewer #1: The authors have addressed my comments. I have no further questions.

Reviewer #2: - This work presents a novel, generalizable metric “reliability score” (or “SRS” in the context of the SWIF(r) classifier), which is applied to diagnose three common issues in machine learning analyses arising from differing contexts between training and testing data (missing classes, cohort effects, missing attributes). Tested on three distinct datasets, the results demonstrate the wide applicability of the SRS and reliability score concept in general at supporting machine learning analyses.

- The authors have painstakingly addressed my previous comments with additional control studies, statistical tests for major results, characterizations of input data, and clarifications in the text. Aside from minor additional contextualization and grammatical fixes, this considerably stronger manuscript is ready for publication in PLOS Computational Biology.

- SRS and systemic bias: The PCA analysis (Fig S6-S7) seems to suggest the existence of clear systemic differences between Male vs. Female cohorts in attribute space, but such differences are not as obviously present when comparing European vs. African cohorts. This result should be used to contextualize why the SRS underperformed in the ancestry mismatch case when compared to the sex mismatch case.

- Introduction and Discussion: Regarding discriminative models, could the SRS still be computed or estimated for a discriminative model that yields class probabilities (possibly by random sampling)? Since any discriminative model can be made to provide class probabilities through ensemble learning, this would significantly increase the generalizability of the SRS.

- Minor Issues

*Figure 2A, Panel 3: Y-axis label “Assymetry” -> “Asymmetry”

*Figure 4B: Legend “Noise Removed Neutral” is missing a colon

*Figure S3-S5, Panel 3: Title “Eliptical” -> “Elliptic”

*Figure S3, Subtitle: Missing period after “Figure S3”

Reviewer #3: The authors have added contextualization and comparisons to existing work as I requested in my review. This analysis shows that SRS performs comparably with other simpler methods. Overall, SRS/SWIF(r) seems like a single framework alternative to an otherwise multi-step process (discriminator + outlier detection), that may not necessarily perform better or faster than the multi-step process. I would suggest that the authors thoroughly discuss the limitations of their approach and address the following points:

- The biggest limitation is that SRS scales quadratically with number of features (n), which the authors haven't mentioned. This is because training and inference of AODE scales quadratically with n. This limits the use of SWIF(r)/SRS to problems with a moderate number of features, and this is reflected in the examples shown in the paper.

- The authors mention "Rather than performing outlier detection or looking for distribution differences prior to performing classification, or as a post-hoc analysis, researchers get SRS output and SWIF(r) classification output in a single step.". This assumes that SWIF(r) is a competitive classification model for any task, which has not been demonstrated in the paper. In many cases, discriminative models like random forests and neural networks would give the best classification performance, and the gap between them and SWIF(r) can be large (the converse has not been demonstrated in the paper). So the claim that researchers can get classification and reliability scores in a single step is not that relevant if the classification performance is sub-par.

- Assuming that outlier detection and classification are performed separately, one can use a cheaper outlier detection method such as LOF (with kd-tree for example) which performs as well as SRS (as per new analyses). In general, for datasets with moderate number of features and examples, it is not that cumbersome to run multiple methods (no timing data provided by the authors). Together, it suggests that one can gain better classification performance and comparable outlier detection using a combination of a discriminative method + LOF (or similar), which would undermine the need to use SRS.

- One of the advantages of SRS posited by the authors is its ability to handle missing data seamlessly. While this is a reasonable advantage, the authors should mention that an alternative to manage missing data is by using imputation methods (e.g. denoising autoencoders) followed by outlier detection methods/discrimanation.

Minor:

- Corrected link that was not working in previous review: https://scikit-lego.netlify.app/mixture-methods.html#outlier-detection. This shows a method very similar to SRS (without class labels).

- "Briefly, from simulated data for each class, SWIF(r) learns a joint, two-dimensional Gaussian mixture for each pair of.." (Page 9)-- "from simulated data" is not a requirement

- I could not find information on how exactly SRS is calculated in the case of missing data. Also how will one account for cases where there are different number of missing variables in different examples?

Reviewer #4: After reading through the authors’ response, their measure is an outlier detection approach that does not fare better than what’s already out there. In addition, while the authors have provided additional information, I do not see convincing evidence is presented that that their approach allows instances from unknown classes and systemic mismatches between training/testing set to be distinguished. My response to the authors’ replies started with “Reviewer response”.

Reviewer #4:

First, there is no question that, when we know the ground truth in either cases, SRS can help

distinguish the presence of an unknown class, or indicate the presence of systemic bias.

However, one of the most challenging issues is that frequently we cannot distinguish 1) [an unknown class present in testing data that was not present in training data], 2) [systemic mismatch between training and testing data]. When we have poorly classified test instances, we almost always do not know if it is a new class or the training data simply cannot lead to a more general model. My understanding is that SRS cannot help us in this regard. So, the claim that SRS can help resolve 1) and 2) only applies

when we know a prior what the situation is, which make it not particularly useful.

We have worked to address these comments in a few ways. First, we think there are

particular instances in which the distribution of the SRS itself can provide some clues. In

particular, the following is an excerpt from The SRS is sensitive to instances from unknown

classes:

“We note the roughly bimodal shape of the SRS scores when a class missing

from training data is represented by many instances in testing data. Combining the

bimodal profile with domain expertise to inspect the low-SRS samples and identify their

common features could lead to the identification of a novel class.”

Reviewer response: Looking into the section where this is explained, it is unclear where the authors present the evidence supporting the above observation. In the sentence prior, Figure S2 is mentioned but it is about a unimodal pattern of SRS score.

In addition, the question of systemic bias is often one that can be plausibly predicted; for

example, in moving from simulated testing data to real-world data, or in trying to use a trained

classifier on a new testing set from a new population or source. In this case, the distribution of

SRS values can serve to assess the generalizability of the trained classifier. The following is an

excerpt from The SRS is sensitive to systemic bias in training data relative to application

data:

“This experiment represents an important use case for the SRS; in many

contexts, researchers may be unsure how well their classifiers will generalize to a new

dataset. The SRS provides a quick way to view the overall difference in model fit

between a training or validation set and a testing set, making it straightforward to test for

significant shifts in SRS distribution using two-sample tests (Rabanser et al. 2019), and

to subsequently make decisions about the need for additional training.”

Reviewer response: The above discussion seems reasonable and would have been convincing should the authors provide evidence that they can readily predict systemic biases with SRS. But this is not presented in the above response.

In situations that do not align with one of the above cases, we agree that this is a fundamentally

difficult problem, and one that really relies on domain-specific knowledge and thoughtful use of

the SRS. We have added text to the Discussion that explores the nuances of this issue:

“One limitation of the SRS is that while the SRS indicates problems with missing

data or deficiencies in the dataset, a solution to those problems may not always be

possible or simple to implement. Researchers will need to rely on domain-specific

knowledge to decide whether the indicated problem can be solved by removing

instances, adding additional classes or attributes to their model, or by including more, or

more diverse instances, in their training data. It may also be advantageous to train

independent classifiers for cohorts that may have distinct underlying properties, for

example, training separate classifiers for each sex in a study where the goal is to

distinguish individuals with a disease phenotype (Mostafavi et al. 2020, Figure 3).

Discovering a solution to low-reliability instances will require answers to subjective and

context-dependent questions that may be difficult to anticipate and may not generalize.

Care should be taken to use abstention, or the removal of instances in a principled and

appropriate manner. When there is risk involved in making the wrong classification,

abstention can be an important practice, as in the medical context (Herbei and

Wegkamp 2009; Gandouz et al. 2021). In other cases there may be a distinctive factor

impacting the performance of the classifier on specific instances: for example, when

simulated training data fails to capture dynamics in unusual genomic regions. Abstention

or removal of instances should not be used to boost measurement of performance of the

classifier, but to invite further investigation and scrutiny. When distribution shifts or

additional classes are suspected, researchers may face decisions about the number and

types of classes to include in training, and cost-benefit considerations when deciding

whether to gather the additional data required to add new attributes or additional training

examples. Increasing transparency and publication of models with negative results

would help address some of the problems raised by these subjective questions [...]

Domain-specific biological knowledge currently provides the best available guide for

navigating the complex, multi-factorial choices involved in model design. In order to best

leverage this domain-specific expertise, we need to ensure that machine learning

models are easy to interpret, dissect and critique, even by researchers without machine

learning expertise.”

Reviewer response: The above detailed discussion is certainly helpful. Nonetheless, my original question remains: where is the evidence supporting SRS’s capability in distinguishing 1) an unknown class present in testing data that was not present in training data, 2) systemic mismatch between training and testing data?

Second, the SRS appears useful in identifying instances that do not resemble training data.

However there are many other measures one can use, including very simple ones such as

Euclidean distance or similarity measures like correlation, to achieve the same goal. SRS’

calculation requires priors and calculation can be involved. Is it better than the other

approaches? If so, can there be some ways to formally demonstrate this?

Thanks to this comment, and similar comments from the other reviewers, we now

include comparisons to other methods that perform outlier or novelty detection. Please refer

here to our response to point #3 from Reviewer #1, which includes references to the new

display items (Figures S2, S3, and S4) in which we compare the SRS to three methods for

novelty detection, as well as excerpts from the Introduction, Results, and Discussion where

discuss the relationship between the SRS and outlier/novelty detection methods, methods for

detection of distribution shift, and the literature about abstention in machine learning classifiers.

Reviewer response: The best practice is to provide response to each reviewer individually as frequently the motivations and suggestions are distinct which is true in this case. In addition, with the very long response document, it is a bit challenging to find what the authors are referring to. In any case, looking into the authors’ response to the other reviewer, they concurred that the approach “has comparable performance”. Thus, while the approach is capable of outlier detection, it does not do worse but not better either. It is a nice feature that SRS can deal with missing data. But apparently this ability does not contribute to better performance. This call into the question the significance of emphasizing SRS’s ability in this regard.

Third, I would love to have some reliability measure for each instance in my classification tasks.

But given the first point, while I appreciate the conceptual basis of a reliability score, SRS

seems to be an outlier detection measure where there are already plenty.

We hope our answers above support the utility of the SRS. In particular, we point to a

couple of excerpts from the Introduction and Discussion where we make this argument:

“By packaging our classifier and the SRS together, we prioritize and highlight the

need for evaluations of outliers and distribution differences to be included in every

machine learning analysis. Rather than performing outlier detection or looking for

distribution differences prior to performing classification, or as a post-hoc analysis,

researchers get SRS output and SWIF(r) classification output in a single step. Exclusion

of outliers and many out-of-distribution instances can be performed by setting a

threshold on the SRS, in much the same way that other methods for outlier and novelty

detection can be applied. The SRS can also be used to measure distribution shifts, fitting

into the “Failing Loudly” framework for machine learning described in Rabanser et al.

2019, in which the authors advocate for machine learning tools that give warnings when

facing unexpected input. In this particular framework, the SRS can serve the role of

dimensionality reduction, a step in the Failing Loudly pipeline prior to testing for

significant distribution shift with subsequent two-sample tests.”

“The reliability score shares some features with outlier detection, novelty

detection and detecting out-of-distribution instances (Rousseeuw and Van Driessen,

1999; Breunig et al. 2000; Liu et al. 2008, Liu et al. 2012, Pimentel et al. 2014, DeVries

and Taylor 2018, Lee et al. 2018), as well as methods for testing distribution shifts

(Rabanser et al. 2019) [...] The SRS is able to operate in the presence of missing

attribute values, a strength that sets it apart from many related methods [...] Across

many areas of biology there is an opportunity to improve machine learning analyses by

creating community standards and best practices for simulation, interpretability and

reproducibility. By packaging SWIF(r) and the SRS together into a single workflow, we

aim to give researchers the tools to both apply the machine learning method, and to

critique and interpret the results they obtain.”

Reviewer response: While a merit of the study as the authors argue is in packaging classification and outlier detection in one setting, one can also argue that such package can be unnecessary since the outlier detection task is not onerous and can be automated easily by any ML practitioner. I see the point about the need to fail loudly but I just am not sure SRS is doing this in ways that the authors have claimed.

**Have the authors made all data and (if applicable) computational code underlying the findings in their manuscript fully available?**

Reviewer #1: Yes

Reviewer #2: Yes

Reviewer #3: Yes

Reviewer #4: Yes

PLOS authors have the option to publish the peer review history of their article (what does this mean?). If published, this will include your full peer review and any attached files.

Reviewer #1: No

Reviewer #2: No

Reviewer #3: No

Reviewer #4: No
---

## [Decision Letter · Decision Letter 2]

10 May 2023

Dear Dr. Sugden,

We are pleased to inform you that your manuscript 'Enabling interpretable machine learning for biological data with reliability scores' has been provisionally accepted for publication in PLOS Computational Biology.

Best regards,

Luis Pedro Coelho

Academic Editor

PLOS Computational Biology

Sushmita Roy

Section Editor

PLOS Computational Biology

Reviewer's Responses to Questions

**Comments to the Authors:**

Reviewer #3: The authors have addressed my comments.

**Have the authors made all data and (if applicable) computational code underlying the findings in their manuscript fully available?**

Reviewer #3: Yes

PLOS authors have the option to publish the peer review history of their article (what does this mean?). If published, this will include your full peer review and any attached files.

Reviewer #3: No

---

## [Editor Report · Acceptance letter]

24 May 2023

PCOMPBIOL-D-22-00274R2 

Enabling interpretable machine learning for biological data with reliability scores

Dear Dr Sugden,

I am pleased to inform you that your manuscript has been formally accepted for publication in PLOS Computational Biology. Your manuscript is now with our production department and you will be notified of the publication date in due course.

With kind regards,

Anita Estes
